DATA RELEASE

# Triatomines outside the Americas: a comprehensive dataset for the global surveillance of Chagas disease vectors

Soledad Ceccarelli[1,2,*], Maria Eugenia Vicente[1,3], Qin Liu[4],
Xiao-Nong Zhou[4], Di Wu[4], Agustin Balsalobre[1,2], Emiliano A. Bruno[1,2],
S. Emilia Barboza[1,2], Romina Valente[1,3] and Gerardo A. Marti[1,2]

1 Centro de Estudios Parasitológicos y de Vectores (CEPAVE-CCT-La Plata-CONICET-UNLP), B1900 La Plata, Buenos Aires, Argentina
2 Consejo Nacional de Investigaciones Científicas y Técnicas (CONICET), C1425FQB Buenos Aires, Argentina
3 CIC, Comisión de Investigaciones Científicas de la Provincia de Buenos Aires, B1900 La Plata, Buenos Aires, Argentina
4 National Institute of Parasitic Diseases, Chinese Center for Disease Control and Prevention, Chinese Center for Tropical Diseases Research, WHO Collaborating Centre for Tropical Diseases, National Center for International Research on Tropical Diseases, Ministry of Science and Technology, Key Laboratory of Parasite and Vector Biology, Ministry of Health, Shanghai, 200025, China

## ABSTRACT

Chagas disease is caused by *Trypanosoma cruzi*, which is transmitted to mammals, including humans, mainly by insects of the subfamily Triatominae (Hemiptera: Reduviidae). Also known as "kissing bugs", the subfamily includes 159 species in 18 genera and five tribes. Although most species are in the Americas, here we present the first compilation of non-American triatomine occurrences. The data (396 records) corresponds to 16 species of the genera *Linschosteus* and *Triatoma* from Africa, Asia, and Oceania collected between 1926 and 2022, and include verified records with geographic coordinates, collection dates, and ecological information. The key novelties of our dataset regard (i) temporal and geographical updates of non-American species, (ii) records of *T. rubrofasciata* hundreds of kilometers inland, and (iii) geographical records of the last two described *Triatoma* species (*T. atrata* and *T. picta*). Our resource supports global surveillance, ecological modeling, and risk assessment by providing evidence of potential vectors for Chagas disease control outside the Americas.

**Submitted:** 06 June 2025

\* Corresponding author. E-mail: soledad.ceccarelli@gmail.com

Preprint submitted at https://doi.org/10.1590/SciELOPreprints.12931

Included in the series: *Vectors of human disease* (https://doi.org/10.46471/GIGABYTE_SERIES_0002)

**Subjects** Ecology, Biodiversity, Global Health

## DATA DESCRIPTION

### Context

Chagas disease, caused by the protozoan *Trypanosoma cruzi* (Chagas, 1909) (NCBI:txid5693) (Kinetoplastida, Trypanosomatidae), is transmitted mainly through the feces of triatomine (Hemiptera: Reduviidae: Triatominae) insect vectors, but may also be transmitted from mother to child, by blood transfusions or infected organ transplants, and by oral transmission through contaminated food and/or beverages. These multiple routes of transmission make Chagas disease an important public health problem, primarily in the

**Figure 1.** Global distribution of people with Chagas disease and triatomine vectors. Green polygons indicate countries with people infected by *Trypanosoma cruzi* according to the last official estimates, 2018 [11]. Orange dots represent records of American triatomine species [9, 10] and red dots represent non-American triatomine species [12, 13].

Americas [1]. However, the migratory movements of people infected with the parasite from the Americas to other continents have contributed to the global spread of *Trypanosoma cruzi* [2], raising the need to strengthen entomological surveillance in regions previously considered non-endemic [3].

Currently, the subfamily Triatominae consists of 156 extant and three fossil species, grouped into five tribes and 18 genera [4, 5]. Since the publication on American triatomine species by Carcavallo *et al.* [6], a new, complete, and integrated database on American triatomine occurrences has been made available [7]. However, since Ryckman and Archbold [8], there has been no integration and updating of information on the distribution of triatomine species outside the Americas. In this context, the main goal of this work is to describe the features of a dataset of non-American triatomine occurrences (henceforth referred to as the "Non-American dataset"), highlighting the most important updates and inclusions. Our Non-American dataset complements the current American triatomine information (henceforth referred to as the "American datasets"), comprising two datasets with more than 35,000 records and made available via the Global Biodiversity Information Facility (GBIF) platform [9, 10] (Figure 1).

This work is the result of an exhaustive review of public information combined with substantial interinstitutional collaborations. This geodatabase may contribute not only to improving knowledge of the biodiversity of the triatomine species outside the Americas, but also to designing improved strategies for health promotion and vector control, with the ability to assess the status and to show the probable impact of Chagas disease management at a global scale.

**Table 1.** Current taxa classification of the Non-American triatomine species (*T. rubrofasciata* is also present in America) according to the last taxonomic classification of Alevi *et al.* [4] and Zhao *et al.* [5].

| Tribes | Genera | Species name (descriptor name(s), year) | Number of records in the dataset |
|---|---|---|---|
| Triatomini | *Linshcosteus* | *carnifex* Distant, 1904 | 1 |
| | | *chota* Lent & Wygodzinsky, 1979 | 1 |
| | | *confumus* Ghauri, 1976 | 8 |
| | | *costalis* Ghauri, 1976 | 9 |
| | | *kali* Lent & Wygodzinsky, 1979 | 4 |
| | | *karupus* Galvao, Patterson, Rocha & Jurberg, 2002 | 2 |
| | | *Linshcosteus sp.* Distant, 1904 | 1 |
| | *Triatoma* | *amicitiae* Lent, 1951 | 1 |
| | | *atrata* Zhao & Cai, 2023 | 3 |
| | | *bouvieri* Larrousse, 1924 | 4 |
| | | *cavernicola* Else & Cheong, 1977 | 2 |
| | | *leopoldi* (Schouteden, 1933) | 6 |
| | | *migrans* Breddin, 1903 | 12 |
| | | *picta* Zhao & Cai, 2023 | 20 |
| | | *pugasi* Lent, 1953 | 1 |
| | | *rubrofasciata* (De Geer, 1773) | 317 |
| | | *sinica* Hsiao, 1965 | 3 |
| | | *Triatoma sp. Laporte, 1832* | 1 |
| | | Total number of records | **396** |

## Non-American triatomine species occurrence dataset

In 1951, Lent assigned the Indo-Pacific species to only two genera, *Triatoma* Laporte, 1832, and *Linshcosteu*s Distant, 1904 [14]. This study integrated data of 16 species from both genera of triatomine (Table 1), increasing to a total of 396 occurrence records. In this dataset, records from the last two described species of *Triatoma* (*T. atrata* Zhao & Cai sp. nov., 2023, and *T. picta* Zhao & Cai sp. nov., 2023 [5]) have been included. These incorporations, together with the 126 species included in the American dataset [9] and the 17 species included in the Argentinean dataset [10], make a total of 158 triatomine species (*T. rubrofasciata* is included in both the new Non-American dataset and the Argentinean dataset due to its global geographic distribution). Except for the particular case of the species *T. rosai* (Alevi, de Oliveira, Caris Garcia, Cesaretto Cristal, Grzyb Delgado, de Freitas Bittinelli, Visinho dos Reis, Ravazi, Bortolozo de Oliveira, Galvão, Vilela de Azeredo-Oliveira and Fernandez Madeira, 2020), which was described in 2020 after the last update of the Argentinean dataset (and therefore not included), occurrence records of all the American triatomine species described up to date [4, 5] are included among the three datasets (American dataset, Argentinean dataset, and Non-American dataset).

The triatomine species included in the Non-American dataset are distributed in 34 countries (or overseas territories) of Africa (*n* = 14, including Réunion (France) and Azores (Portugal)), Asia (*n* = 15), and Oceania (*n* = 5, including Hawaii (United States)) being Indonesia, China, India and Vietnam the countries with the highest amount of species (Table 2). The six species belonging to the genus *Linshcosteus* (*L. carnifex* Distant, 1904, *L. chota* Lent & Wygodzinsky, 1979, *L. confumus* Ghauri, 1976, *L. costalis* Ghauri, 1976, *L. kali* Lent & Wygodzinsky, 1979, and *L. karupus* Galvao, Patterson, Rocha & Jurberg, 2002) have records (*n* = 26) only in India (Table 2).

*Triatoma rubrofasciata* is the only tropicopolitan species of the subfamily Triatominae (Figure 2). It was most frequently found in port cities, although the dataset described here has records in some countries (such as India, China, Vietnam) where this species has been found 100–500 hundreds kilometers inland.

**Table 2.** Triatomine species present in each country or overseas territory.

| Continent | Countries | Species |
|---|---|---|
| Africa | Angola | *Triatoma rubrofasciata* |
| | Central African Republic | *T. rubrofasciata* |
| | Comoros | *T. rubrofasciata* |
| | Democratic Republic of the Congo | *T. rubrofasciata* |
| | France (Réunion) | *T. rubrofasciata* |
| | Madagascar | *T. rubrofasciata* |
| | Mali | *T. rubrofasciata* |
| | Mauritius | *T. rubrofasciata* |
| | Portugal (Azores) | *T. rubrofasciata* |
| | Seychelles | *T. rubrofasciata* |
| | Sierra Leone | *T. rubrofasciata* |
| | South Africa | *T. rubrofasciata* |
| | Sudan | *T. rubrofasciata* |
| | Tanzania | *T. rubrofasciata* |
| Asia | Brunei | *T. migrans* |
| | Cambodia | *T. rubrofasciata* |
| | China | *T. atrata* |
| | | *T. picta* |
| | | *T. rubrofasciata* |
| | | *T. sinica* |
| | Hong Kong | *T. rubrofasciata* |
| | India | *Linshcosteus carnifex* |
| | | *L. chota* |
| | | *L. confumus* |
| | | *L. costalis* |
| | | *L. kali* |
| | | *L. karupus* |
| | | *T. bouvieri* |
| | | *T. migrans* |
| | | *T. rubrofasciata* |
| | Indonesia | *T. leopoldi* |
| | | *T. migrans* |
| | | *T. pugasi* |
| | | *T. rubrofasciata* |
| | Japan | *T. rubrofasciata* |
| | Malysia | *T. cavernicola* |
| | | *T. migrans* |
| | | *T. rubrofasciata* |
| | Myanmar | *T. rubrofasciata* |
| | Philippines | *T. bouvieri* |
| | | *T. migrans* |
| | | *T. rubrofasciata* |
| | Saudi Arabia | *T. rubrofasciata* |
| | Singapore | *T. rubrofasciata* |
| | Sri Lanka | *T. amicitiae* |
| | | *T. rubrofasciata* |
| | Thailand | *T. migrans* |
| | | *T. rubrofasciata* |
| | Vietnam | *T. bouvieri* |
| | | *T. migrans* |
| | | *T. picta* |
| | | *T. rubrofasciata* |
| Oceania | Australia | *T. leopoldi* |
| | Papua New Guinea | *T. leopoldi* |
| | Republic of Kiribati | *T. rubrofasciata* |
| | Tonga | *T. rubrofasciata* |
| | United States (Hawaii) | *T. rubrofasciata* |

The temporal range covered in the dataset is from 1926 to 2022 (Figure 3). Date information was available for 73% of the records (*n* = 289), and 80% of these records (*n* = 239) belong to *T. rubrofasciata*. This species was the first triatomine species to be described in 1773 [15], and the oldest record included in this dataset is from 1963. However, 95.3% of the data of this species have been collected in the last 10 years (Figure 4).



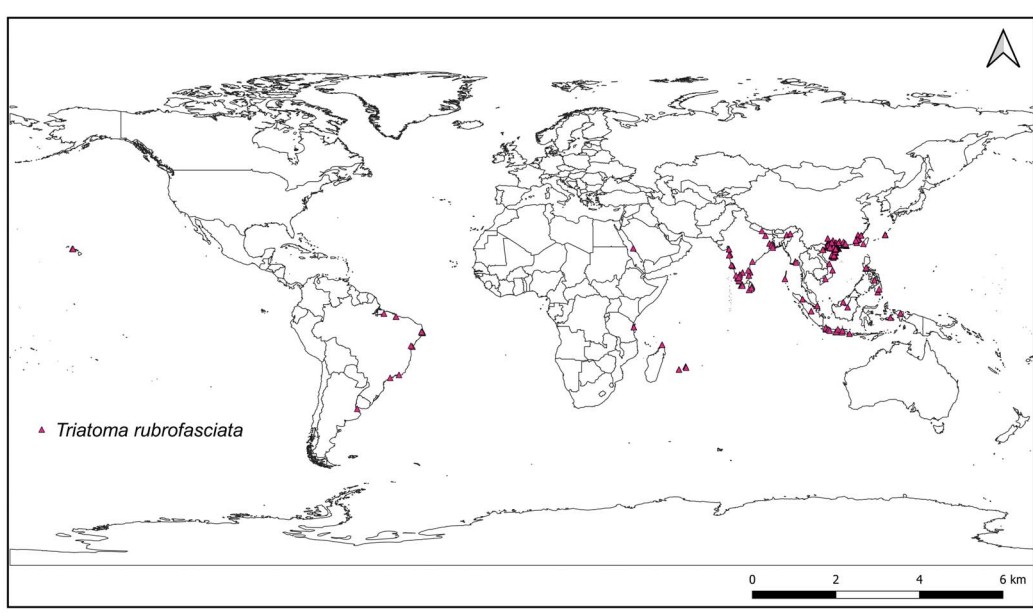

**Figure 2.** Global geographic distribution of *Triatoma rubrofasciata*. Georeferenced occurrences from Africa, Asia, and Oceania belong to the Non-American dataset described here [12] and from GBIF-mediated data [13]. The records from America belong to the Argentinean dataset [10].

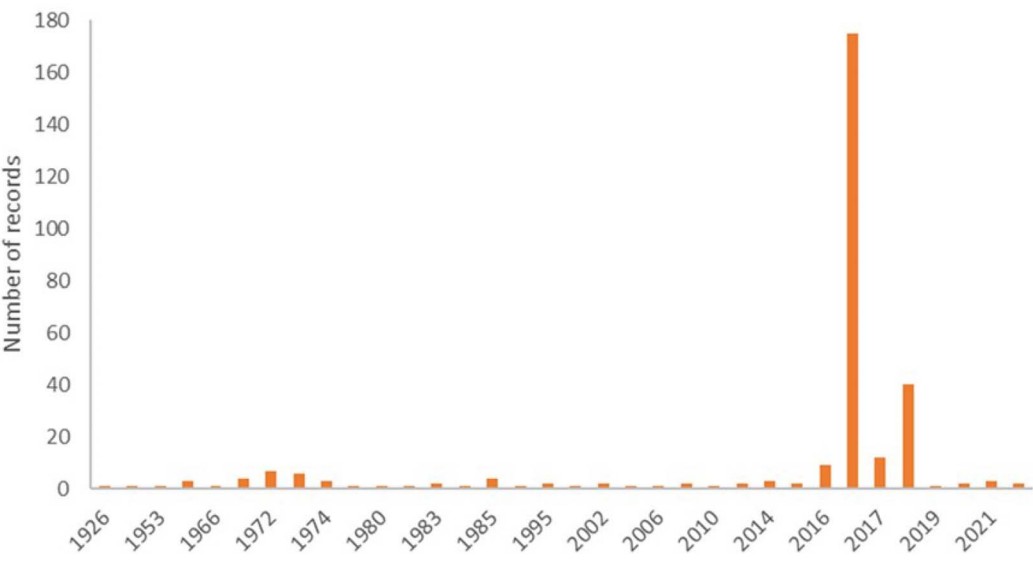

**Figure 3.** Frequency distribution of the number of records per year of all the triatomine species included in the Non-American dataset.

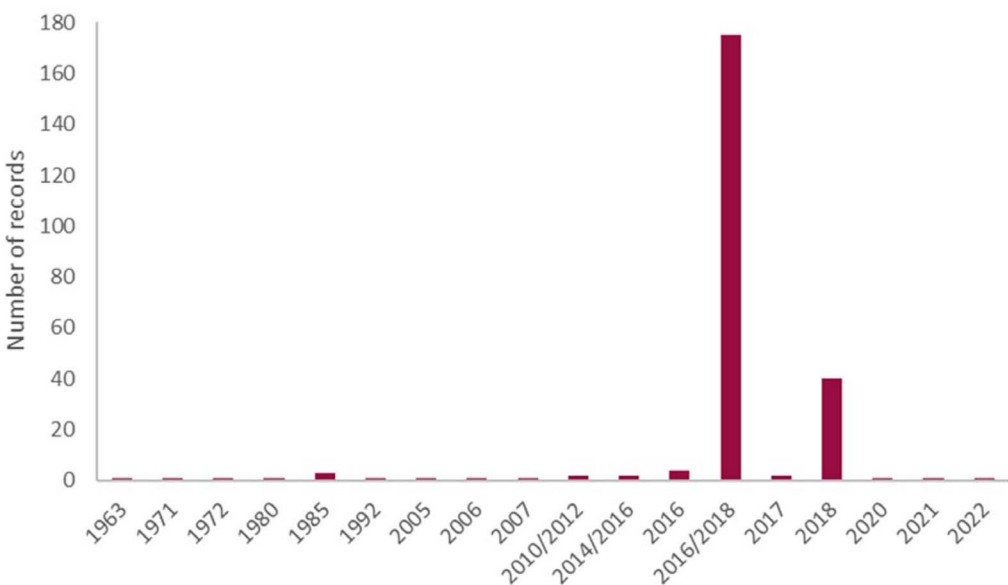

**Figure 4.** Frequency distribution of the number of records per year of *T. rubrofasciata*.

## METHODS

### Information source types and compilation of triatomine species data

To build the dataset, data for each triatomine species were obtained through a detailed and exhaustive review of information. Non-specific temporal range limits were set to obtain the greatest possible amount of new data from as many countries as possible.

Regarding the published information, several public bibliographic online repositories were used (BioOne, Google Scholar, PLoS, PubMed, Scielo, ScienceDirect, and Wiley). They were reviewed using terms such as "Chagas disease" or "Triatominae" plus "Africa", "Asia", and "Oceania" without language restriction.

A large amount of data from China, spanning 2016–2018, was provided by colleagues (co-authors of this work). The geographic information is part of a public paper [16], but the dataset is unpublished.

### Data georeferencing process

To rigorously associate each record with a specific location in the geographical space, data must have information expressed in geographic coordinates (latitude and longitude). If no geographic coordinates were available, the site name was used together with information on administrative divisions to attain an accurate location using Google Earth [17]. Where only the geographic coordinates and the site name (locality) were available, the corresponding administrative divisions were completed using GeoLoc [18]. In the case of records with information only at the state/province level, the geographic coordinates were not added.

The datum (coordinate system and set of reference points used to locate places on Earth) used for all geographic records was the World Geodetic System 1984. The final dataset was built after data quality control.

## Description of the dataset fields

We compiled all relevant and available information associated with each triatomine species. We attached the data to each dataset field, including characteristics of the specimens collected and of the sampled sites. To better describe the fields (based on Darwin Core terms [19]) used to systematize the information, they were grouped into the following six categories: (1) identifiers (including fields used to identify each record, e.g., occurrence ID, institution code, language of the resource, associated references); (2) systematic (including fields used for systematic information, e.g., scientific name, scientific name authorship, taxon rank, and taxon remarks); (3) geographical (including fields with information such as administrative divisions, coordinates, georreference sources); (4) temporal (including fields related to the event date, such as year, month and day); (5) sampling (including fields related to the sampling process, such as name(s) of specimen collector(s), sampled habitat, sampling protocol and effort); and (6) individual (fields related to the total number, sex, and life stage of individuals sampled).

The following subsections provide details about some of the above-mentioned fields requiring specific clarification.

### Systematic fields

When appropriate, the "taxonRemarks" field included notes and/or references about synonyms or formal transferals of the species described in the corresponding record.

### Temporal fields

When a group of specimens' information corresponded to a certain period but with specific available dates, the data were split into different records. If it was not possible to split the data, each record included the original time interval information (in years, months, or days) in the "eventDate" field (e.g., 2016/2018).

### Sampling fields

The "habitat" field refers to the type of habitat where the triatomines were collected, and classified into three categories: domicile, peridomicile, and sylvatic. When specific habitat information was aggregated, the habitat was expressed as a combination of two or three those categories (e.g., domicile–peridomicile, domicile–sylvatic, peridomicile–sylvatic, or domicile–peridomicile–sylvatic).

For the "SamplingProtocols" field, the available information was classified into two major categories: (i) active search, when the searching involved specialized staff; and (ii) passive collection, when different types of traps (e.g., light traps) were used and/or data originated from participative science projects (e.g., iNaturalist).

## DATA VALIDATION AND QUALITY CONTROL

The dataset was subjected to exhaustive quality control. First, each datum was extracted by one person and checked by two others to ensure accuracy and to verify no duplication of records. Subsequently, data were checked to avoid errors (e.g., typing, georeferencing, incorrect locations, synonyms, errors in spelling of administrative divisions) that might have arisen during compilation or data entry. To correct and remove typographical errors and spelling mistakes in the names of administrative divisions, we used OpenRefine software (RRID:SCR_021305) [20], which helps to detect these types of errors in large datasets.



All geographic coordinates were checked using open GIS (QGIS, RRID:SCR_018507 [21]) and Google Earth software [17] to detect georeferenced errors and incorrect locations, ensuring that each point corresponded to a location on the continent and in the correct country. Any outlier coordinates that were geographically distant from the known distribution of a given species were studied to ensure correctness. To detect taxonomic synonym errors, we used the most recent triatomine reviews of currently valid species [4, 5]. If any species name was suspected to be outdated, we consulted the current bibliography or requested the expert opinion of colleagues.

Finally, we improved the quality of our final dataset using the GBIF data validator [22] to identify and address potential issues prior to the dataset publication through the Integrated Publishing Toolkit [23].

## REUSE POTENTIAL

As the information contained within the dataset has been collected using different procedures, this compilation may contain some inherent biases, which should be addressed when the data are used. Most of the data (60%) were obtained from papers published in scientific journals, together with those provided by colleagues (40%). Although data spans 34 countries/overseas territories, China has a volume of data higher than that of other countries; this was due to the significant contribution from colleagues from this country, also co-authors of this work (Qin Liu, Zhou Xiao-Nong, and Di Wu).

Three important notes about the data are: (1) for most species, their presence could be confirmed in countries/states where records had only been available from the 1960s and 1970s; (2) in relation to the above, the GBIF-mediated records (i.e., iNaturalist) used as a complement to show the global distribution of *T. rubrofasciata* (Figure 2) demonstrate the significant contribution for scientific research and policy by current data coming from participatory science projects; (3) making public and available the geographical records from the last two described species of *Triatoma* (*T. atrata* Zhao & Cai sp. nov., 2023 and *T. picta* Zhao & Cai, 2023 [5]).

For habitat sampling, we recognize a potential bias in favor of the domiciliary and peridomiciliary habitats because these are the habitats of greater epidemiological importance. Additionally, the paucity of sylvatic habitat data also results from the difficulty of sampling procedures in the large variety of sylvatic habitats used by triatomines from locations outside those areas where reports are commonly made. Here, we remark on the utility of applications based on citizen-science projects in promoting an increase in reports of sylvatic species. Finally, it is worth noting that about 27% of the records lack available date information, and around 7% of them lack geographic coordinates; thus, we recommend that any analysis based on this dataset should use methods that take such biases into account.

Finally, we would like to remark one reference of a *T. rubrofasciata* record that we considered as having doubtful geographic information (Region of Murcia, Spain) [24], and it was possibly the result of passive transport. Although the record was not included in the dataset, the available evidence indicates this is a plausible mode of transportation for these insects with merchandise. Additionally, it shows that these bugs have high resistance to starvation, and that the highest prevalence of Chagas disease in Europe is in Spain. Hence, we considered that the establishment of a new population of this vector would create a new epidemiological scenario, so we have to be alert.



Despite the information biases described above, the dataset described in this paper, and the complementary American datasets, constitute a valuable compilation of geographic data on American and non-American triatomines, which is as complete, updated, and integrated as possible. Thus, all datasets mentioned herein better represent the number of species and countries, and have more accurate geographic coordinates. Since these datasets are hosted in an open and public repository, we hope that they will contribute to fulfilling national and international goals, such as promoting the exchange of biological information, increasing and improving the accessibility of such information, providing biological data produced and compiled in several countries, and enhancing knowledge of both the biodiversity and epidemiological data related to Chagas disease.

## DATA AVAILABILITY

The dataset "Non-American triatomine occurrence data (Reduviidae:Triatominae)" has been published by Centro de Estudios Parasitológicos y de Vectores (CEPAVE) [25] and is available in the GBIF repository under a CC0 public domain waiver [12]. Additional data is available in GigaDB [26].

## EDITORS' NOTE

This paper is part of a series of Data Release articles working with GBIF and supported by TDR, the Special Programme for Research and Training in Tropical Diseases hosted at the World Health Organization, in order to publish datasets on vectors of human diseases [27].

## DECLARATIONS

### Ethical approval

Not applicable.

### Consent for publication

Not applicable.

### Competing interests

The authors declare that they have no competing interests.

### Authors' contributions

SC, MEV, AB, and RV drafted the data collection protocol. SC, GM, and MEV requested unpublished data from researchers. MEV and SC quality controlled the data; QL, X-NZ, and DW quality controlled the data from China. SC wrote the first draft and all authors contributed to the manuscript. All authors read and approved the final version of the manuscript.

### Funding

This research was funded by grants "Fortalecimiento y promoción de proyectos de ciencia ciudadana" 2022, No 40 (Head Researcher PhD GAM) from Fondo para la Investigación Científica y Tecnológica (FONCYT), Argentina.

### Acknowledgements

The authors are grateful to the people who provided unpublished data, and to the authors who confirmed details related to their published work and who are cited in the published

datasets linked to this data paper. SC and GAM had full access to all the data in the study and take responsibility for the integrity of the data and the accuracy of the data analysis. We thank the Argentine GBIF Node for the valuable technical support provided during the data management, standardization, and publication process.

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
