## [Editor Report]

Editor’s AssessmentTriatomines or kissing bugs are vectors of Chagas disease, caused by the protozoan Trypanosoma cruzi. Chagas disease is predominantly a public health problem in the Americas, however, the increasing risks and migratory movements of people infected with the parasite spreading it to other continents has increased the need to strengthen entomological surveillance in regions previously considered non-endemic. As part of the GBIF and TDR supported vectors of human disease series in GigaByte we have here a dataset of non-American triatomine occurrences. This work being the result of an exhaustive review of public information combined with substantial interinstitutional collaborations (particularly China). In total 396 records were reported between 1926 and 2022, corresponding to 16 species of the genera Linschosteus and Triatoma from Africa, Asia, and Oceania, and include verified records with geographic coordinates, collection dates, and ecological information. Data was validated and peer reviewed, and records that look suspect were fixed or omitted. The dataset described in this paper should constitute a valuable compilation geographic data non-American triatomines, which is as complete, updated, and integrated as possible.Editor’s AssessmentTriatomines or kissing bugs are vectors of Chagas disease, caused by the protozoan Trypanosoma cruzi. Chagas disease is predominantly a public health problem in the Americas, however, the increasing risks and migratory movements of people infected with the parasite spreading it to other continents has increased the need to strengthen entomological surveillance in regions previously considered non-endemic. As part of the GBIF and TDR supported vectors of human disease series in GigaByte we have here a dataset of non-American triatomine occurrences. This work being the result of an exhaustive review of public information combined with substantial interinstitutional collaborations (particularly China). In total 396 records were reported between 1926 and 2022, corresponding to 16 species of the genera Linschosteus and Triatoma from Africa, Asia, and Oceania, and include verified records with geographic coordinates, collection dates, and ecological information. Data was validated and peer reviewed, and records that look suspect were fixed or omitted. The dataset described in this paper should constitute a valuable compilation geographic data non-American triatomines, which is as complete, updated, and integrated as possible.

---

## [Reviewer Report]

Reviewer name and names of any other individual's who aided in reviewer noneDo you understand and agree to our policy of having open and named reviews, and having your review included with the published papers. (If no, please inform the editor that you cannot review this manuscript.)YesIs the language of sufficient quality?YesPlease add additional comments on language quality to clarify if needed
noneAre all data available and do they match the descriptions in the paper? YesAdditional CommentsAre the data and metadata consistent with relevant minimum information or reporting standards? See GigaDB checklists for examples <a href="http://gigadb.org/site/guide" target="_blank">http://gigadb.org/site/guide</a>NoAdditional Commentsbelow in the reviewIs the data acquisition clear, complete and methodologically sound?YesAdditional CommentsIs there sufficient detail in the methods and data-processing steps to allow reproduction?YesAdditional CommentsIs there sufficient data validation and statistical analyses of data quality? YesAdditional CommentsIs the validation suitable for this type of data?YesAdditional CommentsIs there sufficient information for others to reuse this dataset or integrate it with other data?YesAdditional CommentsAny Additional Overall Comments to the AuthorI would like to express my gratitude for the opportunity to review this manuscript. It is a solid, well-written, and highly relevant study for the field of global health, particularly concerning entomological surveillance of Chagas disease in non-endemic regions. The authors’ effort to systematize occurrence data of triatomines outside the Americas bringing together nearly a century’s worth of records into a curated and accessible database represents a significant contribution to the scientific community and public health programs. In the data description, I suggest updating the number of currently recognized genera, which now stands at 19. Please refer to: Paiva VF, de Oliveira J, Belintani T, Galvão C, Gil-Santana HR, da Rosa JA. Hospesneotomae n. gen. of the Triatomini tribe presents a turnaround in the taxonomy of the Triatoma protracta species complex. Sci Rep. 2025 Mar 9;15(1):8143. doi: 10.1038/s41598-025-91399-w. Another relevant point is to consider the occurrence of Triatoma rubrofasciata in Europe: Collantes F, Campos-Serrano JF, Ruiz-Arrondo I. Accidental importation of the vector of Chagas disease, Triatoma rubrofasciata (De Geer, 1773) (Hemiptera, Reduviidae, Triatominae), in Europe. J Vector Ecol. 2023 Jun;48(1):63-65. doi: 10.52707/1081-1710-48.1.63. I would, however, like to suggest a few specific adjustments aimed at improving the clarity and precision of the information presented. Firstly, I noticed a minor inconsistency in the data: the manuscript states that 299 records contain a collection date (“eventDate”), but upon accessing the dataset published on GBIF, I found only 289 records with this field completed. Similarly, it is stated that approximately 33% of records lack a date, whereas the actual figure appears to be closer to 27% (107 records without a date out of 396 total). I recommend reviewing and adjusting these figures to ensure consistency between the manuscript and the dataset. Another important point concerns the dataset’s license. The manuscript refers to a CC0 license, whereas the license registered on GBIF is CC BY-NC 4.0. I suggest correcting this information to accurately reflect the data’s access and reuse policy. I also noted that 46 records include dates that do not follow the ISO 8601 format, which may affect the dataset’s interoperability. I recommend reviewing and standardizing these entries using the YYYY-MM-DD format, in accordance with GBIF guidelines. With respect to the content of the manuscript, I commend the integration of data from citizen science sources (such as iNaturalist), which innovatively broadens the scope and contemporaneity of the dataset. This approach could be further enriched by a brief discussion on the strengths and limitations of such data for entomological surveillance purposes. Finally, I acknowledge and appreciate the transparent discussion of potential biases within the dataset, especially regarding the predominance of records in domestic and peridomestic habitats. It might be valuable to emphasize a bit more the potential impact of this limitation on future ecological analyses and to suggest possible strategies to encourage records from sylvatic environments. In summary, this is an excellent piece of work that, with a few minor adjustments, will certainly have even greater impact and utility. I recommend the acceptance of the manuscript pending minor revisions.RecommendationMinor Revision

---

## [Reviewer Report]

Upload additional filesDRR-202506-01-R01/stage_files/DRR-202506-01/Review MS/DRR-202506-01_Data-Review-BM.pdfReviewer name and names of any other individual's who aided in reviewer Bastien MOLCRETTEDo you understand and agree to our policy of having open and named reviews, and having your review included with the published papers. (If no, please inform the editor that you cannot review this manuscript.)YesIs the language of sufficient quality?YesPlease add additional comments on language quality to clarify if needed
Are all data available and do they match the descriptions in the paper? YesAdditional Comments“Date information was available for 75% of the records (n= 299)”: I only count 289 occurrences with event date in the GBIF dataset; needs to be clarified. Related: “Finally, it is worth noting that about 33% of the records lack available date information”: 107 occurrences missing data info over 396 occurrences = 27% (not 33%)Are the data and metadata consistent with relevant minimum information or reporting standards? See GigaDB checklists for examples <a href="http://gigadb.org/site/guide" target="_blank">http://gigadb.org/site/guide</a>YesAdditional Comments1 - GBIF Non-American dataset catalog number 86 (Azores occurrence) is indicated as continent = Asia, when it is Africa in the manuscript (Africa makes more sense than Asia) 2 - 46 Recorded date are not formatted correctly, need to follow ISO 8601 (see https://discourse.gbif.org/t/please-share-your-dates-correctly/3824)Is the data acquisition clear, complete and methodologically sound?YesAdditional CommentsIs there sufficient detail in the methods and data-processing steps to allow reproduction?YesAdditional CommentsIs there sufficient data validation and statistical analyses of data quality? YesAdditional CommentsIs the validation suitable for this type of data?YesAdditional CommentsIs there sufficient information for others to reuse this dataset or integrate it with other data?YesAdditional CommentsAny Additional Overall Comments to the AuthorNeed to change GBIF dataset license to CC0RecommendationMinor Revision

---

## [Reviewer Report]

Reviewer name and names of any other individual's who aided in reviewer Johan Manuel CalderónDo you understand and agree to our policy of having open and named reviews, and having your review included with the published papers. (If no, please inform the editor that you cannot review this manuscript.)YesIs the language of sufficient quality?YesPlease add additional comments on language quality to clarify if needed
Are all data available and do they match the descriptions in the paper? YesAdditional CommentsAre the data and metadata consistent with relevant minimum information or reporting standards? See GigaDB checklists for examples <a href="http://gigadb.org/site/guide" target="_blank">http://gigadb.org/site/guide</a>YesAdditional CommentsIs the data acquisition clear, complete and methodologically sound?YesAdditional CommentsIs there sufficient detail in the methods and data-processing steps to allow reproduction?YesAdditional CommentsIs there sufficient data validation and statistical analyses of data quality? YesAdditional CommentsIs the validation suitable for this type of data?YesAdditional CommentsIs there sufficient information for others to reuse this dataset or integrate it with other data?YesAdditional CommentsThios dataset is an important contribution to the datasets available for researchers, and they would allow to increase and diversity to amount of information given to ecological models, increasing their prediction power.Any Additional Overall Comments to the AuthorRecommendationAccept